# Human Coronaviruses Do Not Transfer Efficiently between Surfaces in the Absence of Organic Materials

**DOI:** 10.3390/v13071352

**Published:** 2021-07-13

**Authors:** Matthew Dallner, Jennifer Harlow, Neda Nasheri

**Affiliations:** 1National Food Virology Reference Centre, Bureau of Microbial Hazards, Health Canada, Ottawa, ON K1A 0K9, Canada; matthew.dallner@canada.ca (M.D.); jennifer.harlow@canada.ca (J.H.); 2Department of Biochemistry Microbiology and Immunology, Faculty of Medicine, University of Ottawa, Ottawa, ON K1H 8M5, Canada

**Keywords:** human coronavirus 229E, human coronavirus OC43, murine norovirus, transfer, fomite

## Abstract

Human coronaviruses, including SARS-CoV-2, are known to spread mainly via close contact and respiratory droplets. However, other potential means of transmission may be present. Fomite-mediated transmission occurs when viruses are deposited onto a surface and then transfer to a subsequent individual. Surfaces can become contaminated directly from respiratory droplets or from a contaminated hand. Due to mask mandates in many countries around the world, the former is less likely. Hands can become contaminated if respiratory droplets are deposited on them (i.e., coughing or sneezing) or through contact with fecal material where human coronaviruses (HCoVs) can be shed. The focus of this paper is on whether human coronaviruses can transfer efficiently from contaminated hands to food or food contact surfaces. The surfaces chosen were: stainless steel, plastic, cucumber and apple. Transfer was first tested with cellular maintenance media and three viruses: two human coronaviruses, 229E and OC43, and murine norovirus-1, as a surrogate for human norovirus. There was no transfer for either of the human coronaviruses to any of the surfaces. Murine norovirus-1 did transfer to stainless steel, cucumber and apple, with transfer efficiencies of 9.19%, 5.95% and 0.329%, respectively. Human coronavirus OC43 transfer was then tested in the presence of fecal material, and transfer was observed for stainless steel (0.52%), cucumber (19.82%) and apple (15.51%) but not plastic. This study indicates that human coronaviruses do not transfer effectively from contaminated hands to contact surfaces without the presence of fecal material.

## 1. Introduction

The ongoing coronavirus disease-2019 (COVID-19) pandemic is one of the largest pandemics in world history [1]. The cause of this pandemic, severe acute respiratory syndrome coronavirus-2 (SARS-CoV-2) is an enveloped, positive-sense single-stranded RNA virus in the *Coronaviridae* family [2]. *Coronaviridae* is split into four genera: alpha, beta, gamma and deltacoronaviruses. Two of these four genera, alpha and betacoronaviruses, contain species which can infect humans. There are two human coronavirus (HCoV) species in the alphacoronavirus genera, HCoV-229E and HCoV-NL63, both of which cause mild common cold-like infections in humans [3]. Betacoronavirus contains two HCoV species, which also cause mild common cold-like infections in humans, HCoV-OC43 and HCoV-HKU1, but also contains species which cause severe disease. This includes two epidemic causing species, Middle East respiratory syndrome coronavirus (MERS-CoV) and severe acute respiratory syndrome coronavirus-1 (SARS-CoV-1), and the pandemic causing SARS-CoV-2 [3].

Direct transmission from person to person via aerosolized droplets and transmission from direct contact have been shown, and are widely regarded, as the main routes of transmission for SARS-CoV-2 [4,5]. Some concerns have arisen about potential virus transmission via food, food packaging and common food contact surfaces [6]. Fomite-mediated transmission may represent an important secondary route of transmission for SARS-CoV-2, as it has been shown to be a mode of transmission for other viruses, including non-enveloped viruses such as norovirus [7] and rotavirus [8] as well as enveloped viruses including influenza virus [9,10]. Studies have shown that SARS-CoV-2 can remain infectious for 4 to 6 days on plastics and for 3 days on stainless steel [11,12]. Blondin-Brosseau et al. (2020) also showed that human HCoV-229E was able to remain infectious on fresh produce for up to three days, with the survival being produce item dependent, i.e., able to remain infectious on cucumbers for three days but only one day on apples and tomatoes [13]. The ability of HCoVs to survive for multiple days on surfaces would seem to be conducive for fomite-mediated transmission of SARS-CoV-2. However, there has been conflicting evidence in the literature regarding the relative importance of transmission via fomites. Studies from China have suggested that fomite-mediated transmission may have resulted in cases of COVID-19. For example, in Wenzhou, China, after direct contact between individuals was ruled out, researchers suggested that indirect fomite-mediated transmission was occurring [14]. Another example was from Guanzhong, China, where it was determined that a cluster of familial cases was initially caused by an indirect infection of one family member who came into contact with a snot-contaminated elevator button [15]. In contrast to this, however, reports and studies have been published that have shown the risk of fomite-mediated transmission of SARS-CoV-2 is likely to be low [16] and other researchers have also suggested that the risk is likely exaggerated [17,18].

Since mask wearing has been mandated in many countries around the world [19], the direct deposit of respiratory droplets from the mouth and nose onto surfaces is likely to be reduced. Instead, the next most likely scenario is hands contaminated with SARS-CoV-2 depositing the virus onto food and food contact surfaces. Other viruses, including norovirus [20,21], human adenovirus [21] and hepatitis A [22], have all been shown to effectively transfer between contaminated hands and food and food contact surfaces, with transfer efficiencies often being matrix dependent, i.e., better transfer to ham [20] and raspberries [22] than lettuce [20,22]. At the time of writing, there is no literature on the transmission of HCoVs from a contaminated hand to food or food contact surfaces.

While contaminated surfaces may be able to cause an indirect respiratory transmission [23], oral transmission may also occur, followed by an infection in the gastrointestinal (GI) tract which may lead to a subclinical infection in the respiratory tract. The first piece of evidence that supports this mode of transmission is that the cellular receptor for SARS-CoV-2, angiotensin-converting enzyme 2 (ACE-2), is highly expressed in the intestine and colon [24] and studies have demonstrated the ability for HCoVs to survive conditions of the GI tract including the low pH of the stomach, digestive enzymes and bile [25]. Further, gastroenteritis symptoms such as diarrhea and vomiting, are known to be common symptoms in COVID-19 cases, occurring in between 20 and 35% of all cases, and therefore productive infection in the GI tract is likely [26,27]. SARS-CoV-2 is also present, and can be shed, in fecal material [28]. Furthermore, oral inoculation of SARS-CoV-2 in the golden Syrian hamster model resulted in a productive infection in the GI tract and a subsequent subclinical respiratory infection [23].

This study aimed to address the knowledge gap that exists concerning the ability of HCoVs to transfer from contaminated hands to food and food contact surfaces. Further, this study aimed to address whether contamination in the presence of fecal material (representative of fecal viral shedding) could increase viral transfer. Since working with SARS-CoV-2 requires a biosafety level 3 facility, this work was conducted using two other pathogenic HCoVs, HCoV-229E and HCoV-OC43, which were chosen as surrogates to SARS-CoV-2 due to similar physicochemical properties to more pathogenic HCoVs (i.e., MERS and SARS) [29,30]. The use of surrogates allows for expanding the current knowledge of HCoV, without the need for more stringent biosafety measures required to work with the more pathogenic HCoVs. Murine norovirus-1 (MNV-1) was also included in this study to compare enveloped viruses to non-enveloped viruses.

## 2. Materials and Methods

### 2.1. Cell Lines and Viruses

The MRC-5 cell line, human lung fibroblast cells (ATCC#CCL-171) were obtained from the American Type Culture Collection (ATCC). These cells were maintained in complete Eagle’s minimum essential medium (MEM) (Gibco-Invitrogen Co., Grand Island, NY, USA) supplemented with 10% (*v/v*) heat-inactivated fetal bovine serum (FBS) (Gibco-Invitrogen), 1% non-essential amino acids (Gibco-Invitrogen), 1% GlutaMax-1 (Gibco-Invitrogen), 500 µg/mL penicillin/streptomycin (Gibco-Invitrogen), 1% amphotericin B (Gibco-Invitrogen) and 0.22% (*w/v*) sodium bicarbonate (Sigma Aldrich Canada, Oakville, ON, Canada). The BV-2 cell line, mouse microglial cells were obtained courtesy of Dr. Christiane Wobus (University of Michigan, Ann Arbor, MI). These cells were maintained in complete Dulbecco’s modified Eagle medium (DMEM) (Gibco-Invitrogen) supplemented with 10% (*w/v*) heat-inactivated FBS (Gibco-Invitrogen), 500 µg/mL penicillin/streptomycin (Gibco-Invitrogen) and 0.22% (*w/v*) sodium bicarbonate (Sigma Aldrich). Both cell lines were grown at 37 °C and 5% CO_2_ and were split 1:2 or 1:3 every two days using 0.05% Trypsin-EDTA.

The viruses used in this study were HCoV-229E (ATCC#VR-740) and HCoV-OC43 (ATCC#VR-1558), both obtained from ATCC, and MNV-1 (ATCC#VR-1937), which was obtained courtesy of Dr. Herbert Virgin (Washington University School of Medicine, St. Louis, MO, USA). The initial stocks of virus obtained were used in subsequent infections of their host cells to establish a working stock of virus for further experimentation, MRC-5 cells for HCoV-OC43 and HCoV-229E and BV-2 cells for MNV-1.

### 2.2. Surface Preparation

Two produce types and two surface coupons were tested. Royal Gala apples and English cucumbers, obtained from local grocery stores in Ottawa, ON, and two 5 × 5 cm coupons of stainless steel (SS, grade #304) and high-density polyethylene (HDPE, as a representative plastic), which are commonly used in the food industry and packaging, obtained from GlobePharma (GlobePharma Inc., New Brunswick, NJ, USA), were used in this study. The surfaces were prepared, in triplicate for each sample type, by cleaning with a Kim-wipe to remove any dust and then disinfecting with 70% ethanol before patting with a Kim-wipe to dry the surface and further allowing the surface to air dry in a biosafety cabinet to remove any residual ethanol.

### 2.3. Determining Limit of Detection

#### 2.3.1. For HCoV-229E and MNV-1

The limit of detection (LOD) for the HCoV-229E plaque assay after swabbing from cucumbers and apples has been published previously [13]. The LODs for the plaque assay after swabbing from stainless steel and plastic and for all surfaces with MNV-1 were determined as follows. MRC-5 or BV-2 cells were seeded onto 12-well plates and grown for 24 h until 90% confluence was achieved. Serial dilutions of HCoV-229E or MNV-1 were directly spread across a 5 × 5 cm demarcated area of each of the surfaces and allowed to air dry. Using sterile cotton swabs moistened in MEM maintenance media (identical to growth media for BV-2 but with 2% FBS) for HCoV-229E and DMEM maintenance media (identical to growth media for BV-2 but with 0% FBS) for MNV-1, the area was swabbed. After swabbing, the samples were then quantified as follows in Section 2.4.1.

#### 2.3.2. For HCoV-OC43

The limit of detection (LOD) for the HCoV-OC43 TCID_50_ assay after swabbing from all four surfaces was determined as follows. MRC-5 cells were seeded into 96-well plates and grown for 24 h until 90% confluence was achieved. Serial dilutions of HCoV-OC43 were directly spread across a 5 × 5 cm demarcated area of each of the surfaces and allowed to air dry. Using sterile cotton swabs moistened in MEM maintenance media the area was swabbed. The samples were then quantified as follows in Section 2.4.2.

### 2.4. Viral Quantification

Plaque assay was used to quantify HCoV-229E and MNV-1, whereas TCID50 was used to quantify HCoV-OC43. Plaque assays are generally preferred over TCID50 where possible, due to yielding an absolute quantity and being more reliable, although sometimes lacking sensitivity when compared to TCID50 [31]. For HCoV-OC43, plaque assays are not well proven at this time and so TCID50 was used. For means of comparison to plaque assay results, TCID50/mL values were converted to PFU/mL by multiplying by 0.7, which is a constant value obtained based upon Poisson distribution [32].

#### 2.4.1. Plaque Assay for HCoV-229E and MNV-1

Infectious viral particles for HCoV-229E and MNV-1 were determined by plaque assay using MRC-5 and BV-2 cells, respectively, as follows. The cells were grown for 2 to 3 days before being seeded into 12-well plates at a concentration of 5 × 10^5^ cells/well and incubated for 24 h to reach a confluence of 90%. Viral transfer samples were diluted in MEM maintenance media for HCoV-229E and DMEM maintenance media for MNV-1. Once diluted, 100 µL of the dilutions for each of the samples was used to infect wells of the 12-well plate (performed in triplicate) for 90 min at 35 °C and 5% CO_2_ for HCoV 229E and for 60 min at 37 °C and 5% CO_2_ for MNV-1, with shaking every 10 min. A negative control consisting of just MEM/DMEM maintenance media and a positive control consisting of diluted viral stocks were also included. After the infection period was completed, the inocula were removed and the wells were washed once using phosphate-buffered saline (PBS). The cells were then covered with 2 mL of overlay media, which consisted of a 50:50 mix of 2 × MEM growth medium and 0.5% agarose (Gibco-Invitrogen) for HCoV-229E and a 50:50 mix of 2 × DMEM growth media and 2% agarose for MNV-1. Plates were incubated at 35 °C and 5% CO_2_ for 3 days for HCoV-229E and at 37 °C and 5% CO_2_ for 2 days for MNV-1. Plates were then fixed using 3.7% paraformaldehyde for 2 to 4 h. Subsequently, the paraformaldehyde was removed and the overlay plugs were freed using running water. The cells were then stained with 0.1% crystal violet for 20 min and then plaques were counted and PFU/mL determined.

#### 2.4.2. TCID_50_ for HCoV-OC43

Infectious viral particles for HCoV-OC43 were determined by TCID_50_ using MRC-5 cells as follows. The cells were grown for 2 to 3 days before being seeded into 96-well plates at a concentration of 5 × 10^4^ cells/well and incubated for 24 h to reach a confluence of 90%. Viral transfer samples were diluted in MEM maintenance media. Once diluted, 100 µL of the dilutions for each of the samples were used to infect wells of the 96-well plate (performed in quadruplicate). A negative control consisting of MEM maintenance media and a positive control consisting of diluted HCoV-OC43 stock, were also included. The plates were than incubated at 33 °C for 5 days. Next, the media was removed and replaced with 0.1% crystal violet. Wells were then inspected for the presence or absence of visual CPE, and the TCID_50_/mL was calculated using the Reed-Muench method [33]. For means of comparison to plaque assay results, TCID_50_/mL values were converted to PFU/mL by multiplying by 0.7.

### 2.5. Transfer Experiment

#### 2.5.1. Transfer of Viruses to Surfaces Using Maintenance Media as the Transfer Matrix

A 1.0 × 10^6^ PFU/mL dilution of each virus was prepared using their respective maintenance media, MEM maintenance media for HCoV-229E and OC43 and with DMEM maintenance media for MNV-1. A total of 100 µL of this dilution was placed onto the tip of each nitrile-gloved finger (20 µL) for a total concentration of 1.0 × 10^5^ PFU/hand. Gloves were used in place of bare hands as: (I) they are recommended for foodservice workers under certain scenarios; and (II) there are ethical considerations when working with potentially infectious microorganisms on bare hands. The droplets were spread evenly on to the surface of the fingertips and allowed to air dry for 10 min. Once dry, the fingertips were pressed on to the surface of the produce or on to the stainless steel/plastic for 10 s. The surfaces where the fingertips made contact were then swabbed using a sterile cotton swab placed into the respective maintenance mediums. Swabbing of finger pads was also conducted. For this procedure, once the inoculum was dried onto the fingertips, instead of pressing onto the surfaces, the fingertips were swabbed, using an identical procedure used on the surfaces. The samples were then prepared for quantification utilizing plaque assay for HCoV-229E and MNV-1 and tissue-culture infectious dose 50% (TCID_50_) for HCoV-OC43 as mentioned above.

#### 2.5.2. Transfer of HCoV-OC43 to Surfaces Using Organic Fecal Material as the Transfer Matrix

The protocol for this was identical to 2.5.1 with one modification; instead of using maintenance media as the transfer material, fecal material was used. To prepare the transfer material, fecal material, obtained from a healthy donor (10% *w/v*), was mixed with dH_2_O. This was then autoclaved at 121 °C for 15 min. This mixture was then centrifuged at 2500× *g* for 10 min. The supernatant was transferred to a new tube and this was used as the transfer material. HCoV-OC43 was then diluted to 1.0 × 10^6^ in this matrix.

### 2.6. Determining Transfer Efficiency

Transfer efficiency was determined using this equation:(1)$$\%\;transfer\;efficiency=\frac{recovered\;infectious\;viral\;concentration\;from\;surface}{recovered\;infectious\;viral\;concentration\;from\;hands}$$

### 2.7. Statistical Analysis

All statistical analysis was performed using R statistical software [34]. Analysis of variance (ANOVA) was used to determine statistical differences between biological replicates, between different treatments in the BV-2 infectivity assay and between treatments in the binding assays. If ANOVA indicated that there was a significant variance, a Dunnett’s test was completed to determine between which treatments the statistical differences occurred.

## 3. Results

### 3.1. Limit of Detection for Swabbing from Stainless Steel and Plastic

The LOD for all viruses was determined (Table 1). The HCoV-229E LODs for cucumbers and apples were previously determined to be 50 and 125 PFU, respectively [13]. The LODs for stainless steel and plastic were determined in this study to be 53 and 44 PFU, respectively. For HCoV-OC43, the LODs for cucumbers, apples, stainless steel and plastic were 32, 10, 73 and 41 PFU, respectively. For MNV-1, the LODs for cucumbers, apples, stainless steel and plastic were 63, 26, 52 and 30 PFU, respectively.

### 3.2. Viral Transfer Efficiency from Hands to Produce/Surfaces

HCoV-229E, HCoV-OC43 and MNV-1 were inoculated onto gloved hands and, after drying, were pressed onto four different surfaces—stainless steel, plastic, cucumbers and apples. The surfaces were then swabbed and infectious viral transfer determined. Transfer of infectious viral particles was not observed for either of the HCoVs to any of the surfaces tested (Table 2 and Figure 1).

Infectious virus, however, was isolated for both viruses from hand swabbing prior to them being pressed to the surfaces. MNV-1 transfer occurred in 33% of the stainless steel samples, 100% of the cucumber samples, 50% of the apple samples and 0% of the plastic samples (Table 2). The average amount of infectious virus isolated from stainless steel, cucumber and apple (from the samples with successful transfer) was 2.20 ± 0.03, 1.99 ± 0.15 and 0.96 ± 0.31 Log_10_ PFU/mL, respectively (Figure 1). Stainless steel and cucumber transfer was statistically higher than apple, which itself had transfer statistically higher than plastic. Based on these data, transfer efficiencies were determined (Table 3). As no transfer occurred for the HCoVs, the percent transfer efficiency is 0% in all cases. MNV-1 transfer efficiency was highest to stainless steel (9.19 ± 0.68%), followed by cucumber (5.95 ± 2.05%) and apple (0.33 ± 0.3%).

### 3.3. Effect of Fecal Material on Transfer Efficiency of HCoV-OC43

As there was no direct transfer observed for HCoVs, we evaluated the introduction of fecal materials, as a transfer matrix, in the efficiency of viral transfer. For this reason, the transfer experiment was conducted a second time for HCoV-OC43, this time using 10% fecal material as the transfer matrix. The use of fecal material allowed for successful transfer of HCoV-OC43 to three of the four tested surfaces: stainless steel, cucumber and apple, but not to plastic (Table 4 and Figure 2).

Successful transfer occurred in 16.7% of the stainless steel samples, 100% of the apple and cucumber samples and 0% of the plastic samples (Table 4). Cucumber and apple had the highest concentrations of infectious virus transferred, 2.73 ± 0.16 and 2.62 ± 0.16 Log_10_ PFU/mL, respectively, and were not statistically different from each other, followed by stainless steel, 1.17 ± 0.00 Log_10_ PFU/mL, and then plastic, which had no transfer (Figure 2). Transfer, in this case, was most efficient to cucumber (19.82 ± 6.09%) and apples (15.51 ± 6.09%), followed by stainless steel (0.52 ± 0.00%). No transfer to plastic was observed (Table 5).

## 4. Discussion

Fomite-mediated transmission of SARS-CoV-2 has been suggested but has yet to be definitively shown. Fomite transmission occurs when a surface contaminated with infectious virus comes into contact with the nose and mouth of a susceptible individual. During the COVID-19 pandemic, public health measures have been implemented which have placed an emphasis on this mode of transmission, i.e., stringent surface decontamination in community settings and wiping down of food packaging, without a clear understanding of how prevalent this mode of transmission is. As masks are mandated in many countries, the next most likely way a surface would become contaminated is through contact with a contaminated hand. This study aimed to address how efficiently contaminated hands can transfer infectious virus to a surface.

Herein, we have used two common HCoVs as surrogates for SARS-CoV-2—HCoV-293, which is an alphacoronavirus and a betacoronavirus, HCoV-OC43. All HCoVs have similar spike protein structural conformation [35]. However, the receptor-binding domain of SARS-CoV-2 is more similar to HCoV-OC43 than HCoV-229E [35]. Furthermore, it has been demonstrated that SARS-CoV-1 and HCoV-OC43 elicit antibodies that cross-react against related HCoVs [35,36]. Aside from the spike protein, the virion of SARS-CoV-2 and HCoV-OC43 contain sialic acid-binding projections called hemagglutinin-esterase (HE) proteins [37]. These data, together with the fact that HCoV-OC43 is a betacoronavirus, make this virus a better surrogate for highly pathogenic betacoronaviruses such as SARS-CoV-2 and MERS.

Although results may differ between bare hands and gloved hands, gloves were used in this study for a few reasons. This first reason is the ethical considerations of spreading fecal organic material and potentially pathogenic microorganisms onto bare skin. The second is that in food service setting and other public service settings gloves are recommended under certain scenarios [38]. The final reason has been the increase in the use of nitrile gloves in the general population during, and likely in response to, the COVID-19 pandemic [39].

No transfer was observed from a contaminated hand to the food surfaces and fomites used in this study for HCoV-229E and HCoV-OC43 when using maintenance media as the transfer matrix, while MNV-1 did transfer under the same conditions. These findings may suggest that HCoVs transfer poorly via fomites when compared to other viruses, such as MNV-1, particularly when organic material is not present. No detectable transfer was observed for HCoVs compared to a range between 0.33 and 9.19% for MNV-1 in this study. The values for MNV-1 transfer obtained in this study are similar to previous works, i.e., transfer efficiencies to stainless steel, tomatoes and cucumber slices of 0.1, 0.3 and 7%, respectively [21]. Other viruses in similar studies had similar or slightly higher transfer efficiencies, including with feline calicivirus (FCV), another surrogate for norovirus, which had transfer efficiencies from bare fingertips to ham, lettuce and stainless steel of 46 ± 20.3%, 18 ± 5.7% and 13 ± 3.6%, respectively [20], and with hepatitis A virus (HAV), which showed transfer efficiencies from bare fingertips to lettuce of 9.2 ± 0.9% [22]. All three of these viruses (FCV, HAV and MNV-1) are non-enveloped viruses. Non-enveloped viruses are more resistant to various environmental conditions than enveloped viruses such as HCoVs [40]. Because they are more resistant, non-enveloped viruses may have a better ability to remain infectious during drying steps, or during the transfer to surfaces, and may survive longer than enveloped viruses such as HCoVs. This supports the opinion that the fomite-mediated transmission of HCoVs, including SARS-CoV-2, may not be efficient [18]. While cases of COVID-19 have been predicted to be due to fomites, this may more likely be due to direct contamination of these fomites with bodily fluids such as respiratory droplets and oral mucosa [14,40,41], which may aid in protecting the virus. As masks are mandated or recommended in many countries, this scenario would be less likely.

When organic fecal material was used as the transfer matrix instead of maintenance media, HCoV-OC43 transferred successfully to three of four of the surfaces tested. HCoVs are known to be present, and can be shed, in feces of an infected individual. If an individual shedding coronavirus in their feces does not follow proper hand hygiene, it is possible that they could transfer virus to food and fomites. While food is not generally regarded as a source of HCoV infections, fomites are believed to be involved in the transmission of respiratory viruses [40]. Further, studies with SARS-CoV-2 have suggested that oral ingestion may be able to cause an infection [23]. These results indicate that, while respiratory droplets and direct contact are the drivers of cases of COVID-19, contamination of food and fomites with contaminated fecal material needs to be explored further.

Of the four surfaces tested, transfer was either most efficient to stainless steel or cucumber and then typically followed by apple. For MNV-1, transfer was most efficient to stainless steel, while for the HCoV-OC43 organic transfer, it was most efficient to cucumber. For both viruses, transfer to plastic did not occur. Comparing the two produce surfaces, cucumber may support better transfer due its more neutral pH, which could impact viral viability, and its higher surface water activity [42,43]. Comparing the two surface coupons, stainless steel and plastic, the difference in transfer may be due to whether the surface is porous or non-porous, with stainless steel being a non-porous surface and the plastic used in this study, HDPE, being a porous surface. Viruses in general seem to be able to transfer more efficiently to non-porous surface [44] and survive longer on these surfaces [45]. The plastic used in this study, HDPE, is used in different types of food packaging including plastic bottles, food plastic wrap and other clear packaging. Studies have suggested that food packaging was responsible for cases of COVID-19, such as cod packaging [46]. However, this study may indicate, particularly if proper masking procedures are followed, that the risk of transmission of HCoVs via food packaging is minimal.

## 5. Conclusions

The transfer of HCoVs from hands to fomites, including food packaging and foods, is likely to be low and the resulting risk of spreading the virus by this mode will likely be low. However, if the virus is present within organic material, such as fecal material, the transfer becomes significantly more efficient. This interaction should be further explored to investigate the effect of fecal HCoV contamination on viral survival and infectivity. Proper hand hygiene should continue to be followed as it is sufficient, even if fecal contamination were to be present, to prevent this mode of viral transfer.

## Figures and Tables

**Figure 1 viruses-13-01352-f001:**
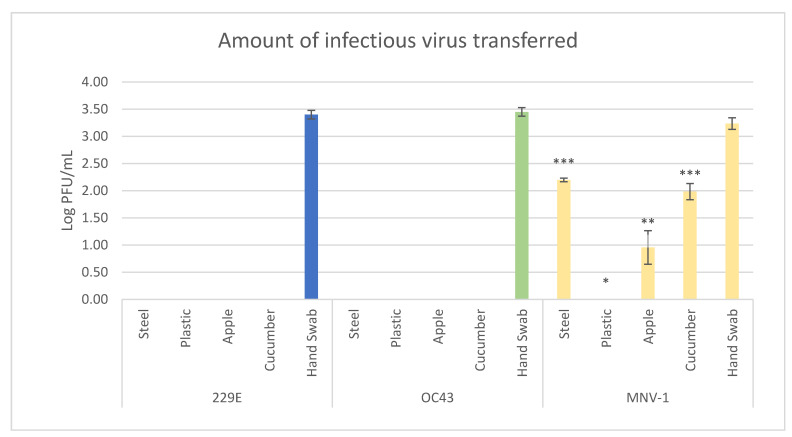
Amount of infectious virus of HCoV-229E, HCoV-OC43 and MNV-1 transferred from artificially contaminated gloved fingertip to the four surfaces tested, and from direct swabbing of fingertips prior to transferring to the surfaces. Different number of asterisks indicate samples which are either not statistically different (same number of asterisks) or are statistically different (different number of asterisks) at *p* = 0.05.

**Figure 2 viruses-13-01352-f002:**
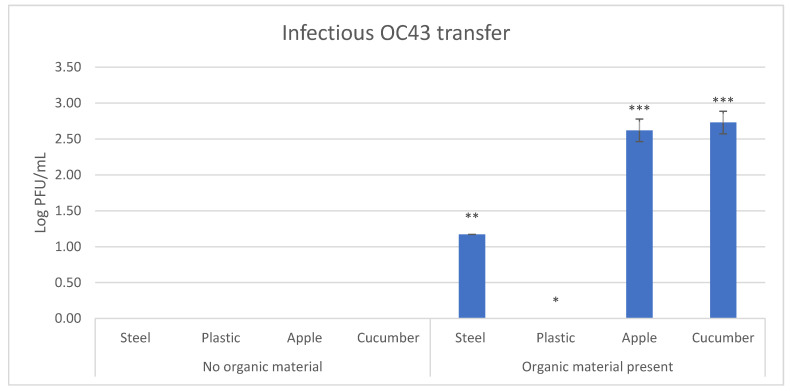
Comparison between infectious HCoV-OC43 transfer with and without the use of organic fecal material to the four surfaces tested. Different number of asterisks indicate samples which are either not statistically different (same number of asterisks) or are statistically different (different number of asterisks) at *p* = 0.05.

**Table 1 viruses-13-01352-t001:** Limit of detection for the detection methods used for the three viruses in this study.

Virus	Surface	Limit of Detection (PFU)
**HCoV-229E**	Stainless steel	53
	Plastic	44
	Apple	125
	Cucumber	50
**HCoV-OC43**	Stainless steel	73
	Plastic	41
	Apple	10
	Cucumber	32
**MNV-1**	Stainless steel	52
	Plastic	30
	Apple	26
	Cucumber	63

**Table 2 viruses-13-01352-t002:** Percent of samples with transfer for HCoV-229E, HCoV-OC43 and MNV-1 to the four surfaces tested.

Virus	Surface	Percent of Samples with Transfer
**HCoV-229E**	Stainless steel	0
	Plastic	0
	Apple	0
	Cucumber	0
**HCoV-OC43**	Stainless steel	0
	Plastic	0
	Apple	0
	Cucumber	0
**MNV-1**	Stainless steel	33
	Plastic	0
	Apple	50
	Cucumber	100

**Table 3 viruses-13-01352-t003:** Percent transfer efficiency for HCoV-229E, HCoV-OC43 and MNV-1 to the four surfaces tested. * = only taking into account samples which had transfer; NT = no transfer.

Virus	Surface	Percent Transfer Efficiency *
**HCoV-229E**	Stainless steel	NT
	Plastic	NT
	Apple	NT
	Cucumber	NT
**HCoV-OC43**	Stainless steel	NT
	Plastic	NT
	Apple	NT
	Cucumber	NT
**MNV-1**	Stainless steel	9.19 ± 0.68
	Plastic	NT
	Apple	0.33 ± 0.03
	Cucumber	5.95 ± 2.05

**Table 4 viruses-13-01352-t004:** Percent of samples with transfer of HCoV-OC43 with or without organic fecal material to the four surfaces tested.

Transfer	Surface	Percent of Samples with Transfer
**No organic material**	Stainless steel	0
	Plastic	0
	Apple	0
	Cucumber	0
**Organic material present**	Stainless steel	16.7
	Plastic	0
	Apple	100
	Cucumber	100

**Table 5 viruses-13-01352-t005:** Percent transfer efficiency of HCoV-OC43 with or without organic fecal material to the four surfaces tested.

Transfer	Surface	Percent Transfer Efficiency *
**No organic material**	Stainless steel	NT
	Plastic	NT
	Apple	NT
	Cucumber	NT
**Organic material present**	Stainless steel	0.52 ± 0.00
	Plastic	NT
	Apple	15.51 ± 6.09
	Cucumber	19.82 ± 6.09

* = only taking into account samples which had transfer; NT = no transfer.

## Data Availability

Data is contained within this article and will be publically accessible upon publication.

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
