# Peer review of "Human Coronaviruses Do Not Transfer Efficiently between Surfaces in the Absence of Organic Materials"

_viruses, 2021, doi:10.3390/v13071352_

Round 1

Reviewer 1 Report

The paper by Dallner et al. addresses the question if human coronaviruses can be transmitted from contaminated hands to food and food contact surfaces. Furthermore, the authors investigated whether presence of fecal material could increase viral transfer (i.e. taking fecal viral shedding into account). For safety reasons they used HCoV-229E and HCoV-OC43 as surrogates to SARS-CoV-2 and compared these enveloped HCoVs to non-enveloped murine norovirus-1 (MNV-1). While the authors observed transfer from MNV-1 from contaminated hands to several surfaces there was no transfer of HCoVs to any of the surfaces. In the presence of fecal material however, HCoV-OC43 transfer was observed for steel, cucumber and apple but not plastic. The authors conclude that (i) HCoVs do not transfer efficiently from contaminated hands to contact surfaces, (ii) contamination of food and fomites with HCoV-containing fecal material needs to be investigated further and (iii) the risk of transmission of HCoVs via food packaging (consisting of plastic) is minimal.

Considering the current SARS-CoV-2 pandemic the paper provides important data concerning the transmission of HCoVs via food and food contact surfaces and therefore I would recommend it for publication. However, I have two concerns, which should be addressed before the paper is published:

  1. The authors use HCCoV-229E and OC43 as surrogates to SARS-CoV-2. They claim that those two viruses have similar physicochemical (“physiochemical” in line 98 is probably a typo) properties when compared to high-pathogenic MERS- and SARS-CoV. In my opinion the authors must be more specific here. Do all viruses have a similar protein content in their envelopes? The amount of protein in the lipid bilayer is critical for the stability of the virions and might impact the ability to transfer to surfaces.
  2. The authors used plaque assays to determine the titers of HCoV-229E and MNV-1 but TCID50 for HCoV-OC43, which makes it difficult to compare the obtained results. Why did they use two different methods? If OC43 does not form plaques, why didn’t the authors use TCID50 for all three viruses?
    The authors convert TCID50 to PFU by multiplying with 0.7 and provide a reference for this method (doi:10.1371/journal.pone.0241022). However, the reference does not explain the factor 0.7 but simply points to another reference which mentions the factor only in its supplementary material without any further explanation. The authors must provide a reference which explains the factor based on the Poisson distribution.

Apart from these two points I think the paper is well written, it presents meaningful experiments, and the results justify the conclusions drawn by the authors.  

Author Response

The paper by Dallner et al. addresses the question if human coronaviruses can be transmitted from contaminated hands to food and food contact surfaces. Furthermore, the authors investigated whether presence of fecal material could increase viral transfer (i.e. taking fecal viral shedding into account). For safety reasons they used HCoV-229E and HCoV-OC43 as surrogates to SARS-CoV-2 and compared these enveloped HCoVs to non-enveloped murine norovirus-1 (MNV-1). While the authors observed transfer from MNV-1 from contaminated hands to several surfaces there was no transfer of HCoVs to any of the surfaces. In the presence of fecal material however, HCoV-OC43 transfer was observed for steel, cucumber and apple but not plastic. The authors conclude that (i) HCoVs do not transfer efficiently from contaminated hands to contact surfaces, (ii) contamination of food and fomites with HCoV-containing fecal material needs to be investigated further and (iii) the risk of transmission of HCoVs via food packaging (consisting of plastic) is minimal.

We would like to thank this reviewer for the thorough review of our manuscript. Below we have addressed the issues raised by this reviewer.

Considering the current SARS-CoV-2 pandemic the paper provides important data concerning the transmission of HCoVs via food and food contact surfaces and therefore I would recommend it for publication. However, I have two concerns, which should be addressed before the paper is published:

  1. The authors use HCCoV-229E and OC43 as surrogates to SARS-CoV-2. They claim that those two viruses have similar physicochemical (“physiochemical” in line 98 is probably a typo) properties when compared to high-pathogenic MERS- and SARS-CoV. In my opinion the authors must be more specific here. Do all viruses have a similar protein content in their envelopes? The amount of protein in the lipid bilayer is critical for the stability of the virions and might impact the ability to transfer to surfaces.

We thank the reviewer for this insightful comment. We corrected the typing mistake and to address this comment regarding the physicochemical properties of human coronaviruses, we have added these lines to the discussion (L276-284):

“Herein, we have used two common HCoVs as surrogates for SARS-CoV-2. HCoV-293, which is an alphacoronavirus and a betacoronavirus, HCoV-OC43. All HCoV have similar spike protein structural conformation [34], however, the receptor-binding domain of SARS-CoV-2 is more similar to HCoV-OC43 than HCoV-229E [34]. Furthermore, it has been demonstrated that SARS-CoV-1 and HCoV-OC43 elicit antibodies that cross-react against related HCoVs [34,35]. Aside from the spike protein, the virion of SARS-CoV-2 and HCoV-OC43 contain sialic acid-binding projections called hemagglutinin-esterase (HE) proteins [36]. These data, together with the fact that HCoV-OC43 is a betacoronavirus, make this virus a better surrogate for highly pathogenic betacoronaviruses such as SARS-CoV-2 and MERS.”

  1. The authors used plaque assays to determine the titers of HCoV-229E and MNV-1 but TCID50 for HCoV-OC43, which makes it difficult to compare the obtained results. Why did they use two different methods? If OC43 does not form plaques, why didn’t the authors use TCID50 for all three viruses?
    The authors convert TCID50 to PFU by multiplying with 0.7 and provide a reference for this method (doi:10.1371/journal.pone.0241022). However, the reference does not explain the factor 0.7 but simply points to another reference which mentions the factor only in its supplementary material without any further explanation. The authors must provide a reference which explains the factor based on the Poisson distribution.

We do agree with the reviewer that including two different methods for quantification of infectious viral particles might be confusing. However, this is due to the limitations that exist for quantification of these viruses. Plaque assay remains the “gold standard” and was employed for quantification of HCoV-229E and MNV, however, there is no reliable and robust plaque assay method for HCoV-OC43. Since including HCoV-OC43was important as a betacoronavirus surrogate, we used TCID50 as the “next best method” for quantification of infectious virus and we included a direct reference for the conversion to plaque forming unit. Therefore, we added these lines to the Material and Methods (L160-166):

“Plaque assay was used to quantify HCoV-229E and MNV-1, whereas TCID50 was used to quantify HCoV-OC43. Plaque assays are generally preferred over TCID50 where possible, due to yielding an absolute quantity and being more reliable, although sometimes lacking sensitivity when compared to TCID50 [31]. For HCoV-OC43 plaque assays are not well proven at this time and so TCID50 was used. For means of comparison to plaque assay results, TCID50/mL values were converted to PFU/mL by multiplying by 0.7, which is a constant value obtained based upon Poisson distribution [32].”

Reviewer 2 Report

The research is highly relevant to be published. However, while the study is aiming at answering the question "whether human coronaviruses can transfer efficiently from contaminated hands to food or food contact surfaces", such was not convincingly shown. The issue is that transfer was studied from nitrile gloves to food and food contact surfaces. That is essentially different, because affinity of the virus for the skin surface may be different from the affinity of the virus for nitrile surface. In other words, bare hand or fingertips would probably provide another outcome of the study.  Somehow, this needs to be dealt with in the manuscript. 

Author Response

We do agree with the reviewer that the affinity of the virus would be essentially different between nitrile gloves and bare hands and thus the viral transfer could be different as well. However, we chose gloved hands for two main reasons: 1) we were interested in studying viral transfer by food handlers in ready-to-eat factory settings or in restaurants, where gloves are often mandatory 2) putting infectious human coronavirus on bare hands (during a pandemic that was caused by a coronavirus), would cause ethical complications and would make finding volunteers challenging. Therefore, to address this comments we added these lines to the Materials and Methods (L201-203):

 “Gloves were used in place of bare hands as they: (i) are recommended for foodservice workers under certain scenarios; and (ii) the ethical considerations when working with potentially infectious microorganisms on bare hands.”

Also added these lines to the Discussion (L285-291):

“Although results may differ between bare hands and gloved hands, gloves were used in this study for a few reasons. This first reason is the ethical considerations of spreading fecal organic material and potentially pathogenic microorganisms onto bare skin. The second is that in food service setting and other public service settings gloves are recommended under certain scenarios [37]. The final reason has been the increase in the use of nitrile gloves in the general population during, and likely in response to, the COVID-19 pandemic [38].”

Round 2

Reviewer 2 Report

The revised manuscript dealt in a satisfactory way with my concerns related to "gloves were used in place of bare hands".